# Analysis of a Capacitive Sensing Circuit and Sensitive Structure Based on a Low-Temperature-Drift Planar Transformer

**DOI:** 10.3390/s22239284

**Published:** 2022-11-29

**Authors:** Yanlin Sui, Tao Yu, Longqi Wang, Zhi Wang, Ke Xue, Yuzhu Chen, Xin Liu, Yongkun Chen

**Affiliations:** 1Changchun Institute of Optics, Fine Mechanics and Physics, Chinese Academy of Sciences, Changchun 130033, China; 2School of Electronic Information Engineering, Changchun University of Science and Technology, Changchun 130022, China; 3School of Fundamental Physics and Mathematical Sciences, Hangzhou Institute for Advanced Study, University of Chinese Academy of Sciences, Hangzhou 310024, China

**Keywords:** capacitive sensing, planar transformer, gravitational wave detection

## Abstract

In space gravitational-wave-detection missions, inertial sensors are used as the core loads, and their acceleration noise needs to reach $3\times 10^{-15}\;{\mathrm{ms}}^{-2}/\sqrt{\mathrm{Hz}}$ at a frequency of 0.1 mHz, which corresponds to the capacitive sensing system; the capacitive sensing noise on the sensitive axis needs to reach $1\;\mathrm{aF}/\sqrt{\mathrm{Hz}}$. Unlike traditional circuit noise evaluation, the noise in the mHz frequency band is dominated by the thermal noise and the 1/f noise of the device, which is a challenging technical goal. In this paper, a low-frequency, high-precision resonant capacitor bridge method based on a planar transformer is used. Compared with the traditional winding transformer, the developed planar transformer has the advantages of low temperature drift and low 1/f noise. For closed-loop measurements of capacitive sensing circuits and sensitive structures, the minimum capacitive resolution in the time domain is about 3 aF, which is far lower than the scientific measurement resolution requirement of 5.8 fF for gravitational wave detection. The capacitive sensing noise is converted to $1.095\;\mathrm{aF}/\sqrt{\mathrm{Hz}}$ in the frequency band of 10 mHz–1 Hz. Although there is a gap between the closed-loop measurement results and the final index, the measurement environment is an experimental condition without temperature control on the ground; additionally, in China, the measurement integrity and actual measurement results of the capacitive sensing function have reached a domestic leading level. This is the realization of China’s future space gravitational wave exploration.

## 1. Introduction

In space gravitational-wave-detection missions, inertial sensors are used as the core load, and their accuracy levels directly limit the sensitivity level of the final low-frequency band of the space gravitational-wave-detection mission. The sensitive structure and capacitive sensing circuit are the core components of an inertial sensor, and the sensitive structure includes the test mass and the electrode cage. Capacitive sensing circuits measure changes in capacitance caused by the movement of the test mass in sensitive structures, and its measurement accuracy has a direct impact on the resolution of the sensor. According to the requirements of the top-level indicators of the detection system, its acceleration noise needs to reach $3\times 10^{-15}\;{\mathrm{ms}}^{-2}/\sqrt{\mathrm{Hz}}$ at a frequency of 0.1 mHz, which corresponds to the capacitive transmission on the sensitive axis of the capacitive sensing system. Sensing noise needs to be as low as $1\;\mathrm{aF}/\sqrt{\mathrm{Hz}}$. The performance of capacitive sensing directly affects the accuracy and even the success or failure of the space gravitational-wave-detection system. Different from the traditional idea of circuit signal-to-noise ratio optimization, in the mHz frequency band, the thermal noise and the device’s 1/f noise is dominant, and circuit noise optimization for low-frequency bands requires new design ideas. Therefore, it is of great scientific significance to carry out research on the scheme of high-precision capacitive sensing systems [1,2,3,4,5].

Because the bridge-detection circuit has the advantages of high resolution, high stability, and not being easily affected by parasitic capacitance, it has been successfully applied to precision measurement instruments, such as inclinometers and gravimeters, from as early as 1973. Transformers are used to convert the displacement of the mechanical structure into an electronic signal. The electrostatic accelerometer series products used in gravity satellites developed by ONERA in France and the space inertial sensors in the LISA program developed by the University of Trento in Italy all use bridge-type capacitive sensing detection circuits based on traditional winding planar transformers [6]. In 2017, the LISA team reported the performance of the capacitive sensing system of inertial sensors on the LISA Pathfinder spacecraft [7]. Through measurements during spaceflight, the system has demonstrated a capacitive sensing noise level of $1\;\mathrm{aF}/\sqrt{\mathrm{Hz}}$, matching the performance metrics of the LISA Pathfinder, and is poised to be applied to subsequent LISA gravitational-wave-detection projects. The experiments carried out by the relevant domestic research institutions on the capacitive sensing system have been carried out on the self-made capacitive calibration platform [8], but the capacitive calibration platform is not the actual measurement object in actual inertial sensors. There are also capacitive sensing systems applied to accelerometers [9], and the resolution used to satisfy them is $2\times 10^{-12}{\;\mathrm{ms}}^{-2}/\sqrt{\mathrm{Hz}}$, the design ideas of accelerometers and inertial sensors are different, and the accelerometer does not include detection of angular acceleration.

The input signal of capacitive sensing passes through the transformer, and the signal-to-noise ratio at the resonant point of the transformer determines the signal accuracy level of capacitive sensing in the low-frequency band. Traditional transformers are wound with copper wires, which are easily affected by temperature, and it is difficult to use them to meet the capacitive sensing index in the millihertz frequency band. Traditional transformers are wound with copper wires. The planar transformer circuit uses PCB wiring instead of copper wires, which has the advantages of low temperature drift and low 1/f noise; however, the design of planar transformers is complicated, often requiring the introduction of interleaved capacitors due to improper design [10,11]. This leads to a reduction in the transformer quality factor (Q value), which is unable to meet the index requirements. So far, research on planar transformers used in capacitive sensing systems at home and abroad has not reached deep enough.

Capacitive sensing systems can not only be used in the detection of gravitational waves in space [12,13,14], but can also be used in the precise measurement of inertial drag effects, in testing the equivalent principle [15,16,17], and in measuring relativistic effects with gyroscopes [18] and other applications.

Based on the requirements of China’s future space gravitational-wave-detection Taiji program, for the low-frequency, high-precision resonant bridge circuit, a capacitive sensing circuit and a closed-loop measurement tool for sensitive structures are developed for a planar transformer with high Q value, low temperature drift, and low 1/f noise. The minimum capacitance resolution in the time domain is about 3 aF, which is much lower than the scientific measurement resolution requirement of 5.8 fF for gravitational wave detection. In the frequency band of 10 mHz–1 Hz, the capacitive sensing noise is measured to be $1.095\;\mathrm{aF}/\sqrt{\mathrm{Hz}}$. Although there is a gap between the closed-loop measurement results and the final index, the measurement environment is an experimental condition without temperature control on the ground, and the measurement integrity and actual measurement results of the capacitive sensing function have reached the domestic leading level in China; this is the realization of China’s future space gravitational wave exploration. It is an important technology that forms a design-and-evaluation principle for low-frequency, high-precision capacitive sensing systems.

The content of this article is as follows. Section 2 introduces the design of the planar transformer, and the composition and basic working principle of the capacitive sensing circuit. Section 3 presents the ground experimental setup and actual measurements, as well as data evaluation results. The final section provides a comprehensive summary of the work of this paper.

## 2. Working Principle of Capacitive Sensing Circuits

As shown in Figure 1, when the distance between the equivalent test mass (TM) and the two electrodes is equal—that is, when the distance between the two electrodes and the TM is d—the effective area of the electrode plate is S. Capacitance values with the same capacitance between the two electrodes and the TM, denoted as $C_{1}$ and $C_{2}$, are as follows:(1)$$C_{1}=C_{2}=\frac{\varepsilon_{r}\varepsilon_{0}S}{d}$$
where $\varepsilon_{0}$ is the vacuum permittivity and $\varepsilon_{r}$ is the relative permittivity.

The displacement of the test mass will cause a change in the differential capacitance, and the TM will produce a small displacement; Δd; $C_{1}$ and $C_{2}$ will accordingly change.
(2)$$\Delta C=C_{1}-C_{2}=2\varepsilon_{r}\varepsilon_{0}S\frac{\Delta d}{d^{2}}$$

In this paper, the capacitive sensing readout circuit of the capacitive sensing system adopts a bridge-detection circuit scheme, which converts the differential capacitance into a DC voltage value and outputs it. Based on a capacitor–inductor resonant bridge, the capacitor, $C_{p}$, and the transformer inductance, L, are designed to resonate at a specific frequency of 100 kHz. An excitation signal with a frequency of 100 kHz was injected onto the TM through a set of six injection electrodes (brown electrodes in Figure 2).

Figure 2 is a block diagram of the single TM-sensing channel electronics along the *x*-axis, with the remaining channels omitted for simplicity. Two pairs of electrodes (A+/A−, B+/B−) allow simultaneous measurement of TM displacement and rotation by measuring the gap between the electrodes on opposite sides of the TM. For small TM displacements, the capacitance is proportional to the corresponding TM electrode gap. An AC-excitation signal is injected into the TM. The differential current that is amplified and converted to an AC-induced voltage proportional to the TM displacement is filtered at the AC-excitation frequency, and the final amplitude is demodulated and converted to a DC voltage digital value. The capacitive sensing system circuit includes an analog front-end and an AC-amplifier circuit, a band-pass filter circuit, a demodulation circuit, an analog-to-digital conversion circuit, and a total of four functional module circuits.

Similarly, the other eight electrodes (not shown) were used to derive TM motion. The TM is surrounded by four sense/drive electrodes on each axis. A stable 100 kHz AC signal was applied to the TM through the red excitation-injection electrode in Figure 1. The electrode pair A+/A− is defined as the first sensing channel of the *x*-axis, and the electrode pair B+/B− is defined as the second sensing channel of the *x*-axis. To overcome the need to specify different position sensing requirements for each axis (displacement or rotation), the requirement is to convert to capacitance and have an equal range on all axes (channels), i.e., scientific measurement resolution of 5.8 fF. The range is ±0.12 pF for the mode and ±2.5 pF for the wide-range (WR) mode. The capacitive sensing noise of the key technical indicators is required to be measured in the scientific measurement (HR) mode, so this paper focuses on the scientific measurement (HR) mode.

### 2.1. Equivalent Sensitive Structure for Torsion Balance

Equivalent test masses for torsion balances are at the heart of mechanically sensitive structures. It is an equivalent cube with a side length of 46 mm and a mass of about 107 g, made of platinum, gold, and aluminum alloys.

The equivalent test mass (Figure 3) is surrounded by an electrode housing that houses 12 sensing/driving electrodes (2 electrodes on each side—green electrodes in the figure), forming 6 pairs of sensing electrodes.

Six excitation electrodes (one each on the Y side of the equivalent test mass and two on each Z side—brown electrodes in the figure) are used to inject excitation signals.

### 2.2. Analog Front-End and AC-Amplifier Circuit

As shown in Figure 4, the analog front-end and AC-amplifier circuits are composed of tuning capacitors, transformers, preamplifiers, and main amplifiers. The purpose of the preamplifier is to receive and amplify the differential transformer signal. The main advantage of the transimpedance amplifiers with the gain controlled by a single component (a small feedback capacitor in the pF range) is that the transfer function is very flat around the resonance point, which makes the design less sensitive to temperature changes. Therefore, the transimpedance amplifier was chosen as the design solution for the preamplifier. In addition, for the sake of symmetry and high SNR, the design scheme of the differential transimpedance amplifier can be adopted.

The differential capacitance change is the amount of positional change in the test mass. The excitation signal, $V_{m}$, is injected into the differential capacitor, and the voltage generated at the amplifier end after the transformer is calculated as follows:(3)$$U_{op}=\frac{S^{2}L\left(C_{1}-C_{2}\right)}{1+S^{2}L\left(C_{1}+C_{2}+C_{p1}+C_{p2}\right)}V_{m}\;=V_{m}\frac{s^{2}L}{1+s^{2}LC_{eq}}\Delta C$$
where $C_{eq}=2\left(C_{0}+C_{p}\right)$, $C_{0}$ is the capacitance when the TM is at the center of the electrode. The value of $C_{0}$ can be calculated by Equation (3), where the tuning capacitor $C_{p}=C_{p1}=C_{p2}$ and the transformer inductance value is L. Equation (3) shows that the angular frequency of the transfer function consisting of the differential transformer and capacitor is ω.
(4)$$\omega =\frac{1}{\sqrt{LC_{eq}}}$$

When the angular frequency of the input signal is ω, the circuit will resonate with the input signal, and the output voltage and impedance will both reach their maximum values. In the design of this scheme, the resonant frequency is set to 100 kHz by matching the capacitor parameter, $C_{eq}$.
(5)$$U_{op}=V_{m}\frac{-\omega^{2}L}{1-\omega^{2}LC_{eq}}\Delta C$$

The voltage noise amplitude spectral density [18] of the capacitive sensing transformer bridge is (unit $V/\sqrt{\mathrm{Hz}}$) [19]:(6)$$S_{V}^{\frac{1}{2}}=\sqrt{4KT\omega_{0}L\frac{1+Q^{2}}{Q}}=\sqrt{4KT\omega_{0}LQ}$$
where *Q* is the transformer quality factor, $\omega_{0}$ is the 100 kHz angular frequency, $V_{m}$ is the excitation signal, *K* is the Boltzmann factor, and *T* is the temperature.

The excitation signal, $V_{m}$, is injected into the differential capacitor, and the voltage/tolerance gain value (unit V/F) generated by the amplifier end of the transformer is calculated as follows:(7)$$\left|\frac{\partial U_{op}}{\partial \Delta C}\right|=\left|j\omega Z_{op}(\omega_{0})\right|V_{m}=\omega_{0}^{2}LQV_{m}$$
where Q is the transformer quality factor, the minimum value of $V_{m}$ is 0.43V, and $\omega_{0}$ is the 100 kHz corner frequency.

### 2.3. Planar Transformer Design

The bridge output is an amplitude-modulated signal with the carrier frequency $\omega_{0}$ that is demodulated and this changes the noise density. The demodulation process doubles the noise power spectral density (PSD).
(8)$$S_{out}\left(\omega_{LF}\right)=S_{in}\left(\omega_{0}\right)$$
where “LF” stands for a low-frequency band of interest ($f_{LF}$ < 1 Hz). It is clear from (8) that the ASD of the voltage and capacitance noise after demodulation is larger by a factor of $\sqrt{2}$ compared with before demodulation. Define LQ as the inductance quality factor, τ, of the resonant point of the transformer. According to Formulas (6)–(8), the equivalent capacitive sensing noise of the thermal noise of the transformer bridge can be written as:(9)$$S^{\frac{1}{2}}=\frac{1}{V_{m}}\sqrt{\frac{8KT}{\omega_{0}^{3}LQ}}$$

The space gravitational wave detection index requires the capacitive sensing noise to be better than $1\;\mathrm{aF}/\sqrt{\mathrm{Hz}}$, and τ needs to be greater than 0.74 H. Therefore, we need to design a transformer with a large inductance (4.2 mH) and a high *Q* value (180). The transformer parameters are substituted into the analog front-end and the AC-amplifier circuit to simulate the voltage noise and frequency response curves. The results are shown in Figure 5.

The voltage noise is the lowest point of noise at a frequency of 100 kHz, and the voltage noise is $3\;\mu V/\sqrt{\mathrm{Hz}}$. According to Formula (6), 3.504 V/pF is obtained; that is, the equivalent capacitive sensing noise of the thermal noise of the transformer bridge is $0.857\;\mathrm{aF}/\sqrt{\mathrm{Hz}}$, meeting the index requirements.

We choose planar transformers to replace the more common traditional wound transformers. Traditional wound transformers are made of copper wire by hand, so the production is simple, the processing cycle is short, and the cost is low, but the disadvantage is that the copper wire is easily affected by heat, and same-batch transformers have poor parameter consistency. Planar transformers use PCB internal wiring and winding, so the transformer parameters produced in the same batch are consistent, accurate, and controllable, and the PCB thermal expansion coefficient is low.

Designing a planar transformer requires consideration of core selection, PCB board selection, and winding design. According to the permeability coefficient and temperature coefficient of the magnetic core at the operating frequency of 100 kHz, we use manganese–zinc ferrite N48 [20] as the magnetic core of our planar transformer. According to the inductance coefficient of N48 ($A_{L}=630\;\mathrm{nH}$) and the required transformer inductance value (4.2 mH), we calculate the number of winding turns as follows:(10)$$A_{L}=\frac{L}{N^{2}}$$
(11)$$\sum \frac{l}{A}=\frac{l_{e}}{A_{e}}$$
(12)$$L=\frac{\mu_{e}\mu_{0}N^{2}}{\sum \frac{l}{A}}$$

It is calculated that the number of winding turns N=80 is a reasonable design value.

The board needs to have a relatively low dielectric constant to minimize interleaving capacitance, and multilayer boards need to be fabricated to meet the number of turns required. This is divided into three planar winding boards (two primary and one secondary) by the available depth based on the chosen core N48 size. To reduce staggered capacitance, two annular spacers are made of the same substrate. The required 80 winding turns are divided into sixteen layers of circuit boards to achieve five turns per layer, and the designed inductance value is generated. A deconstructed view of a planar transformer is shown in Figure 6.

Each board has five turns of winding copper wire, and the top and bottom layers of each laminate are slightly staggered by about 0.1 mm to reduce stagger capacitance. Each laminate will connect the upper set of five turns to the lower set via blind buried vias. The two-primary planar winding circuit and the two-secondary planar winding circuit are identical because of the good repeatability between all windings in printed circuit board technology, which fully reflects the design advantages of planar transformers. Figure 7 shows the planar transformer developed in this paper.

### 2.4. Band-Pass Filter Module, Demodulation Module, and Analog-to-Digital Conversion Module

The band-pass filter module selects the frequency of the signal output by the AC-amplifier module and filters out the frequencies other than the excitation signal. The band-pass filter in this paper uses a low-pass filter in series with a high-pass filter, and finally adjusts the amplitude to ensure that the output amplitude remains unchanged. As shown in Figure 8, the bandwidth range is 71~139 kHz. Since the demodulation module is not ideal, large out-of-band noise may leak into the demodulated frequency band. Therefore, for a narrower bandpass filter, the design difficulty of the demodulation module will be reduced. On the other hand, the gain of a filter with a very narrow passband is more sensitive to changes in its components (resistors and capacitors) with temperature, for which the above bandwidth range is chosen. The parameters of the band-pass filter are summarized in Table 1 and shown in Figure 8.

The function of the demodulation module is to complete the lock-in amplification and to extract the modulated AC information in the AC signal by adjusting the phase of the demodulation clock. This text adopts the scheme of switch demodulation. The modulated signal first needs to go through the electronic switch demodulation circuit to separate the 100 kHz carrier signal from the noise spectrum and to separate the voltage signal close to DC. The AC signal is averaged to a DC voltage and an analog switch is used to rectify the signal. Since the noise is randomly phase-shifted relative to the demodulator control signal, the rectified noise averages zero.

After the signal goes to DC, the low-frequency drift of the op amp (OP) can greatly affect the sensing resolution. Therefore, all the OPs in the circuit use auto-zero amplifiers, which basically do not exhibit 1/f noise characteristics. The post-stage adopts a third-order Butterworth low-pass filter, and the −3 dB corner frequency is located at 5 Hz. The final stage consists of non-inverting and inverting auto-zero buffers, each including low-pass filtering to eliminate auto-zero switching noise.

The low-noise analog-to-digital conversion module uses the analog-to-digital conversion chip to convert the analog quantity of the capacitive sensing circuit into a digital quantity and enter the digital control circuit for processing. AD7712 (Analog Devices, Wilmington, MA, USA) is a complete analog front-end chip suitable for low-frequency measurement applications. It has 24-bit high-precision analog-to-digital conversion performance and has two analog input channels that can directly receive low-level signals from sensing circuits. The AD7712 has an internal PGA gain function that can be configured from 1× to 128× the analog gain. The working-principle diagram of the analog-to-digital conversion module is shown in Figure 9.

The electronic design parameters of the capacitive sensing system are shown in Table 2.

## 3. Experiment and Calibration

### 3.1. Planar Transformer

As shown on the left side of Figure 10, the LCR meter (E4980AL, Keysight, Santa Rosa, CA, USA) is used to measure the parameters of the planar transformer. The inductance value, L, is 4.3 mH, the quality factor, Q, is 196, and the measured value of LQ is 0.85 H, which meets the design requirements of 0.74 H. As shown on the right side of Figure 10, the planar transformer is brought into the capacitive sensing circuit system of this paper for experiments.

### 3.2. Capacitance Measurement and Capacitance Sensing Circuit Gain Calibration

The measured object of capacitive sensing is the same equivalent test mass and electrode cage as in Figure 1: the sensitive structure. The capacitive sensing circuit and the sensitive structure form a closed-loop measurement. The test environment is shown in Figure 11. The position information of the test mass is reflected in the tolerance value of the six pairs of electrodes. The sensitive structure is fixed on the hexapod PI console, and the hexapod PI console can be controlled to adjust the position of the test mass within the sensitive structure. The sensitive structural parameters are shown in Table 3.

In this experiment, the AH2700A ultra-precision capacitor bridge is used to measure the capacitance tolerance value of each electrode pair (Figure 3—brown electrode). The negative terminal of the AH2700A ultra-precision capacitor bridge is connected to X. A+, X. B+, and six other measurement-sensitive structures have differences in the electrode tolerance; the measured voltage values are shown in Table 4.

In fitting the circuit gain curve, as shown in Figure 12, we find that the circuit gain is 19.14 V/pF, the full range of scientific measurement (HR) mode is ±0.12 pF, and the full-scale tolerance corresponds to ±2.3 V voltage value, which meets the AD chip ±2.5 V range. By controlling the hexapod PI console to uniformly adjust the position of the test mass, the tolerance values of multiple sets of *X*-axis directions are obtained by measuring with a capacitance meter, which corresponds to the voltage output value obtained by the capacitance sensing and electrostatic servo control subsystem. The capacitive sensor’s DC voltage output reading, fitted to a circuit gain curve, is consistent with Figure 12.

### 3.3. Gain Calibration of Each Functional Module of Capacitive Sensing

According to the gain curve of the fitting circuit, the overall gain of the measured capacitive sensing system is 19.14 V/pF.

The internal PGA gain of AD7712 is set to four times gain: the gain from the analog front-end AC-amplifier circuit to the demodulation output is 4.785 V/pF.

As shown in Figure 13, the input is 100 kHz, the sinusoidal signal to the demodulation module is 408 mV(RMS); an oscilloscope MSOS404A was used to measure the demodulation output, 501 mV(RMS), and the DC voltage, and it calibrates the demodulation circuit to a gain of 1.23 times. According to the gain of the demodulation circuit, it can be calculated that the gain from the analog front-end AC-amplifier circuit to the output of the band-pass filter is 3.89 V/pF.

A vector network analyzer E5061B (Keysight, Santa Rosa, CA, USA) is used to measure the input and output gain of the band-pass filter circuit function module, as shown in Figure 14. The vector network analyzer measurement result is 1.1766 dB at 100 kHz—1.144 times the gain.

Figure 15 shows the various gains for the capacitive sensing circuit. According to the gain of the band-pass filter circuit, it can be calculated that the gain of the analog front-end AC amplifier circuit is 3.40 V/pF.

### 3.4. Calculation of Minimum Resolution of Capacitive Sensing

By controlling the hexapod PI console to evenly adjust the position of the test mass, the tolerance value in the *X*-axis direction is measured with a capacitance meter, so that the tolerance is 0.14 fF, and the capacitive sensing analog front-end is connected to the spectrum analyzer DSA705 (Rigol, Suzhou, China). The spectrum analyzer measurement results are shown in Figure 16.

In the spectrum analyzer 50 ohm impedance system, under the input condition of 0.137 fF tolerance, the output measurement result of the analog front-end circuit at 100 kHz frequency is 467 uV, and the gain of the analog front-end AC-amplifier circuit calculated before is 3.40 V/pF, which is consistent with the input tolerance.

Connect the analog front-end AC-amplifier circuit input to ground, and connect the capacitive sensing analog front end to a spectrum analyzer.

The spectrum analyzer measurement results are shown in Figure 17. In a spectrum analyzer 50 ohm impedance system, with the input grounded, the output of the analog front-end circuit at 100 kHz measured −87 dBm under a voltage of 10 uV. The link gain between the analog front-end and the AC amplification part is 3.40 V/pF, which is the minimum capacitance resolution of 2.93 aF, and the noise is the lowest at 100 kHz.

## 4. Measurement Results and Discussion

The capacitive sensing system connects the A+ and A− and the B+ and B− electrodes in the *X*-axis direction of the electrode cage of the sensitive structure in the following connection mode, and measures for 80 h. The temperature stable area is intercepted, and the time-domain voltage values of the *X*-axis A+ and A− electrodes are shown in Figure 18. When there is no excitation input, the time-domain jitter of the AD terminal reading is about 60 uV, 19.14 V/pF, and the minimum capacitance resolution in the real-time domain is about 3 aF, which is consistent with the measurement results in Section 3.4; the measurement results are far below the scientific measurement resolution of 5.8 fF required for gravitational wave detection.

We use a four-wire PT1000 platinum resistance to measure the temperature of the capacitive sensing electronics. The data-collection model is DAQ970A. There is no temperature-control measure for the ambient temperature. As shown in Figure 19b, at the beginning of the experiment, the capacitive sensing system starts cold. Because there is no temperature-control measure in this experiment, there is an obvious temperature rise and a stable trend in the first half of the figure. The DAQ970A measures the single-point temperature change in the experimental environment. The resistance–capacitance and op amp devices on the sensing circuit need a longer time to achieve temperature balance. When the measurement time is long enough, the temperature and the capacitive sensing time-domain voltage tend to be stable at the same time, and the capacitive sensing time-domain voltage has no obvious temperature drift.

As shown in Figure 20, in the capacitive sensing noise spectral density, the PSD square root of the selected *X*-axis channels A+ and A− is shown by the basic spectral change, shown by the blue curve, but the spectral resolution of this method is low, and there are many big distortions. At the same time, we performed a power spectral density (PSD) linear fit to the capacitive sensing time-domain voltage noise data, which is represented by the blue curve. The red curve is calculated using Welch’s average periodogram method [21]. The data were cut into overlapping segments of $4\times 10^{4}$ s and filtered with a Blackman–Harris window of a minimum of four samples [22]. All data were processed using the LISA Technology Package for Data Analysis (LTPDA) toolbox [23,24]. Welch’s average periodogram method can reduce random fluctuations, has better convergence, and has more accurate power spectral density performance.

According to the capacitive sensing noise graph in Figure 20, it is lower than $1.095\;\mathrm{aF}/\sqrt{\mathrm{Hz}}$ in the range of 10 mHz–1 Hz. This experiment is a ground experiment, and the electric box has not taken any temperature-control measures, so the frequency band that the attention index meets is 10 mHz–1 Hz. In Table 5, the research status of capacitive sensing at home and abroad is compared and explained. The capacitive sensing system developed by us is applied to inertial sensors, but the capacitive sensing system developed in reference [9], and the acceleration noise index requirement is $2\times 10^{-12}\;{\mathrm{ms}}^{-2}/\sqrt{\mathrm{Hz}}$.This study is different from the application of the capacitive sensing system developed in reference [9]. The inertial sensor not only has the function of an accelerometer, but also includes the measurement of angular acceleration and multi-degree-of-freedom motion, so the sensitive structure of the two is different from the relevant electronic parameters (such as the amplitude $V_{m}$ of the excitation signal). The capacitive induction noise in Figure 20 is the closed-loop test result of the capacitive sensing system and the sensitive structure of the inertial sensor, and the index of comparison corresponds to an acceleration noise of $3\times 10^{-15}{\;\mathrm{ms}}^{-2}/\sqrt{\mathrm{Hz}}$. Additionally, we introduce the minimum capacitive resolution method in the capacitive sensing time domain for the first detailed analysis.

## 5. Conclusions

The capacitive sensing circuit presented in this paper adopts a low-frequency, high-precision resonant capacitive bridge method based on a planar transformer. We introduce the minimum capacitive resolution method in the capacitive sensing time domain for the first detailed analysis; the minimum capacitance resolution in the time domain is about 3 aF, which is much lower than the scientific measurement resolution of 5.8 fF required for gravitational wave detection. It is converted into capacitive sensing noise in the frequency band of 10 mHz–1 Hz, and the measured value is $1.095\;\mathrm{aF}/\sqrt{\mathrm{Hz}}$. The low-frequency noise contribution comes from temperature drift and the 1/f noise of the device. Although there is a gap between the closed-loop measurement results and the final index, the measurement environment is the experimental condition without temperature control on the ground; that is, the low-frequency test results of capacitive sensing noise still include the noise caused by temperature drift, which also shows that the use of the planar transformer-based, low-frequency, high-precision resonant capacitor bridge method can effectively suppress low-frequency 1/f noise. For the present capacitive sensing system, there is still room for improvement in the low-frequency noise performance. Subsequent related experiments will consider the temperature-control measures of the whole box. A real-time demodulation-phase-monitoring system will also be investigated in the future for the next iteration of the capacitive sensing electronic system. The measurement integrity and actual measurement results of the capacitive sensing function in this paper have reached the domestic leading level in China, enabling an important technology for China’s future space gravitational wave exploration.

## Figures and Tables

**Figure 1 sensors-22-09284-f001:**
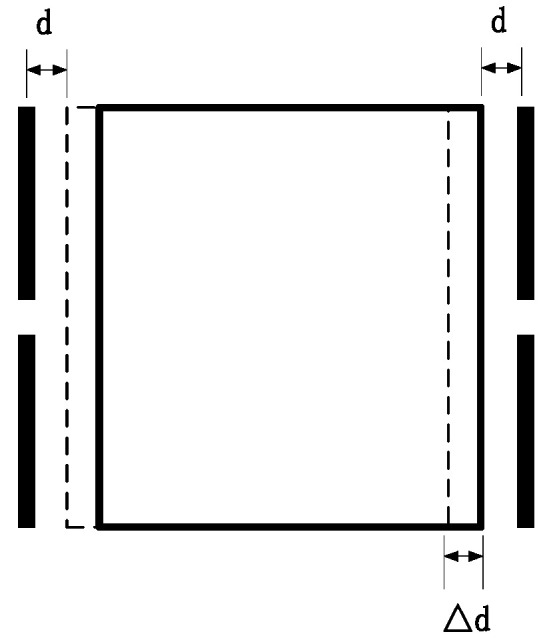
Schematic diagram of the equivalent test mass and the distance between the two electrodes being equal.

**Figure 2 sensors-22-09284-f002:**
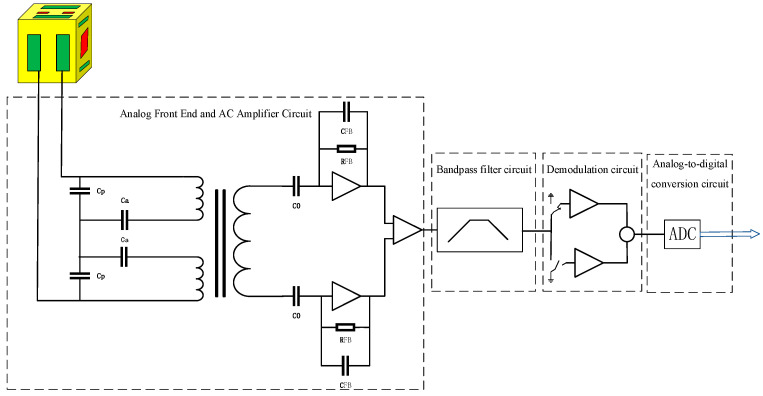
Block diagram of the single TM-sensing channel electronics along the *x*-axis.

**Figure 3 sensors-22-09284-f003:**
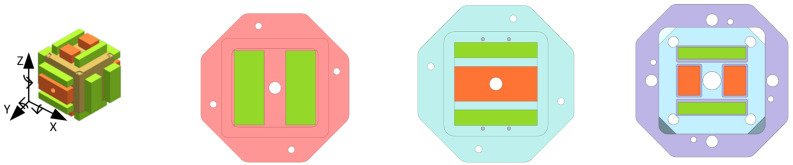
Yellow represents the test mass; green represents the sensing/driving electrode; brown represents the excitation-injection electrode.

**Figure 4 sensors-22-09284-f004:**
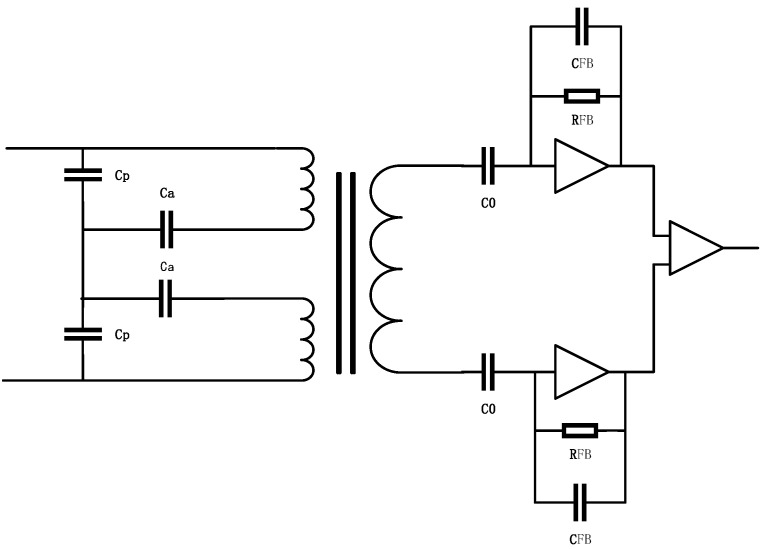
Analog front-end and AC-amplifier circuits.

**Figure 5 sensors-22-09284-f005:**
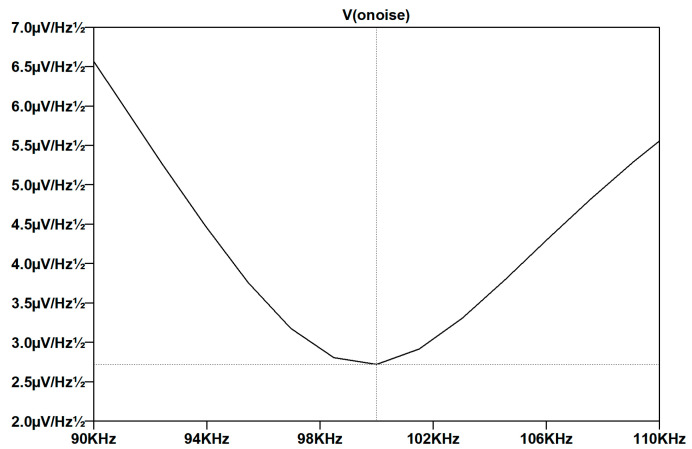
Voltage noise vs. frequency response curve.

**Figure 6 sensors-22-09284-f006:**
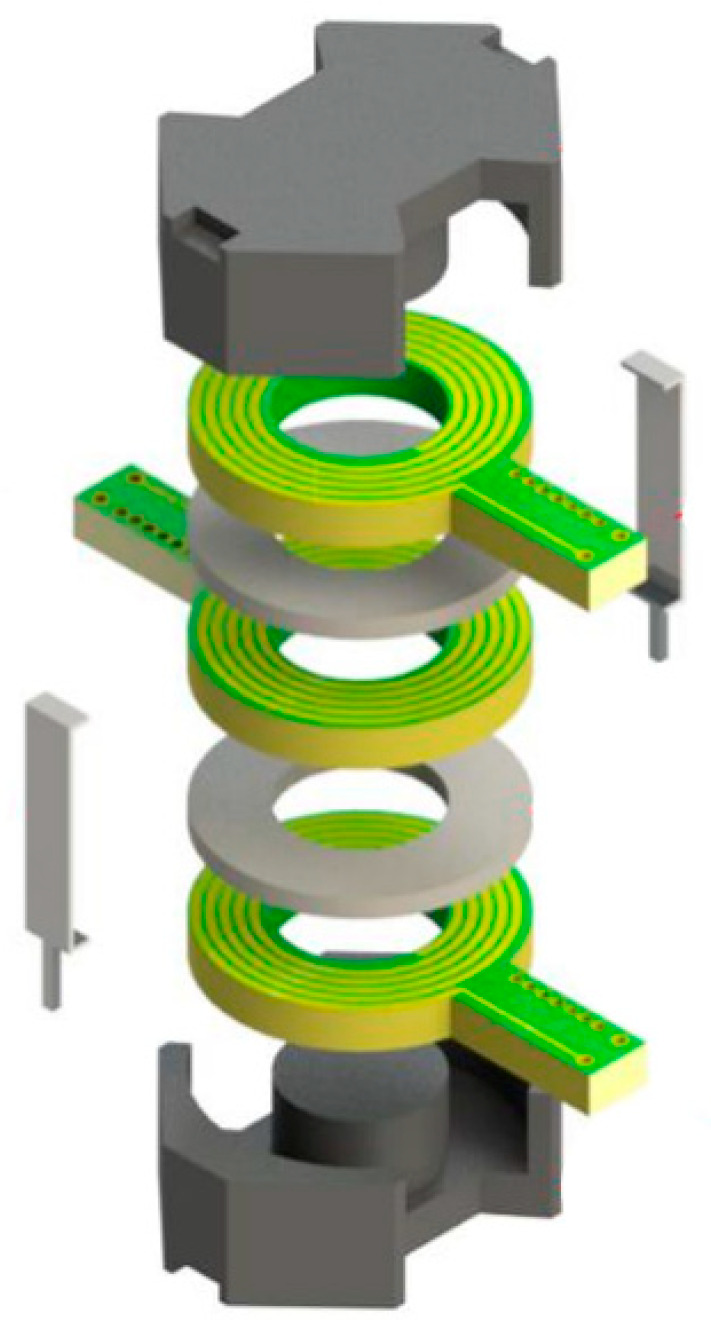
Deconstructed view along the ferrite core, planar winding circuit board, spacers, and clamps.

**Figure 7 sensors-22-09284-f007:**
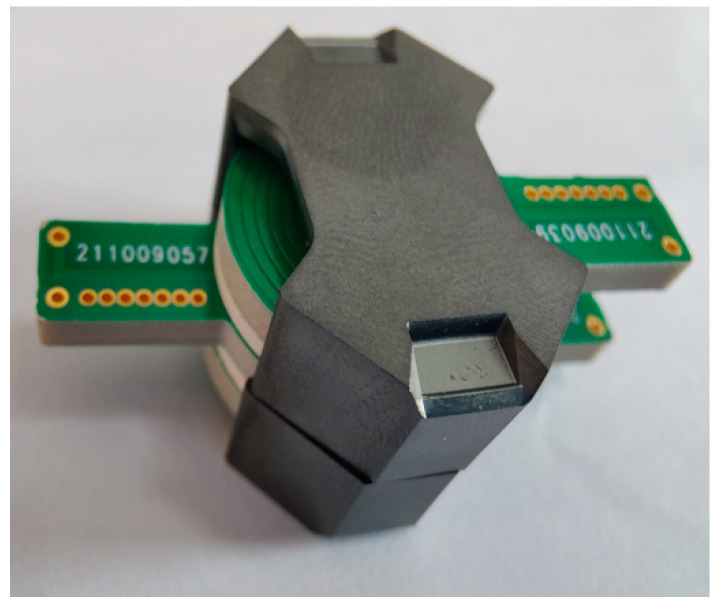
Planar transformer developed in this paper.

**Figure 8 sensors-22-09284-f008:**
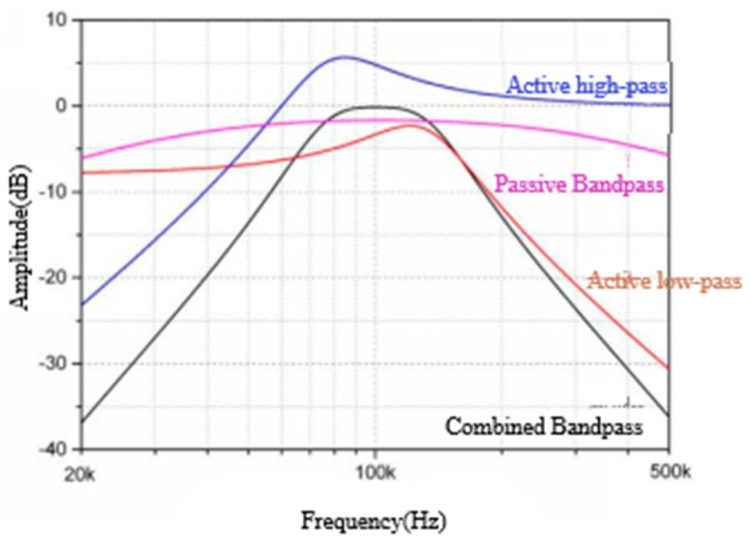
Amplitude–frequency characteristic curve of the band-pass filter.

**Figure 9 sensors-22-09284-f009:**
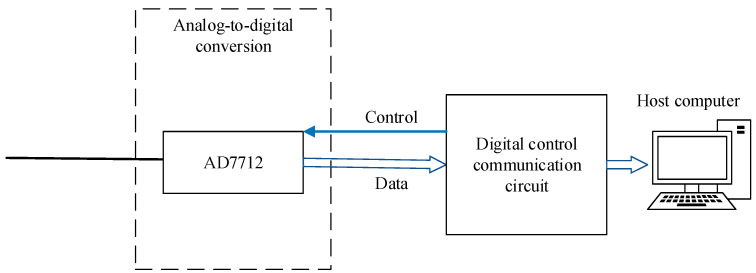
Working-principle diagram of analog-to-digital conversion module.

**Figure 10 sensors-22-09284-f010:**
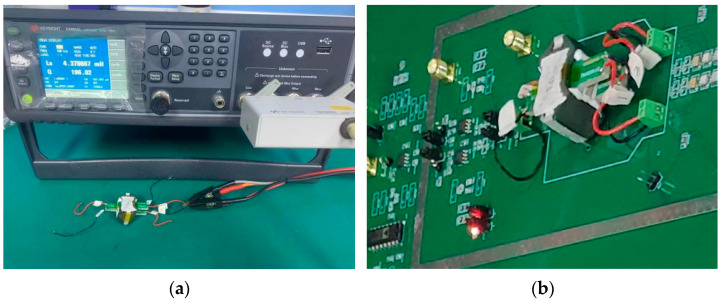
(**a**) LCR meter used to measure the parameters of the self-developed planar transformer; (**b**) the experiment of bringing the planar transformer into the capacitive sensing circuit system.

**Figure 11 sensors-22-09284-f011:**
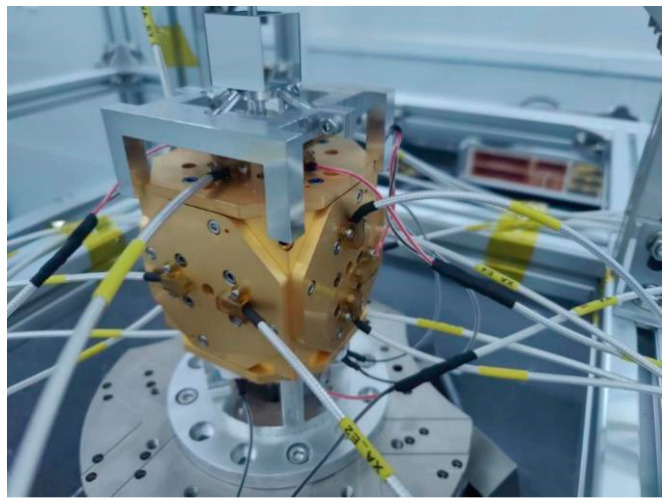
Physical map of the ground test of sensitive structures.

**Figure 12 sensors-22-09284-f012:**
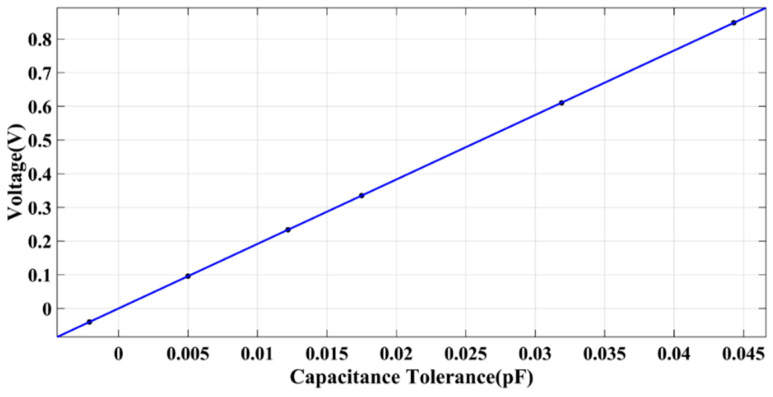
Fitted circuit gain curve for the capacitive sensing system.

**Figure 13 sensors-22-09284-f013:**
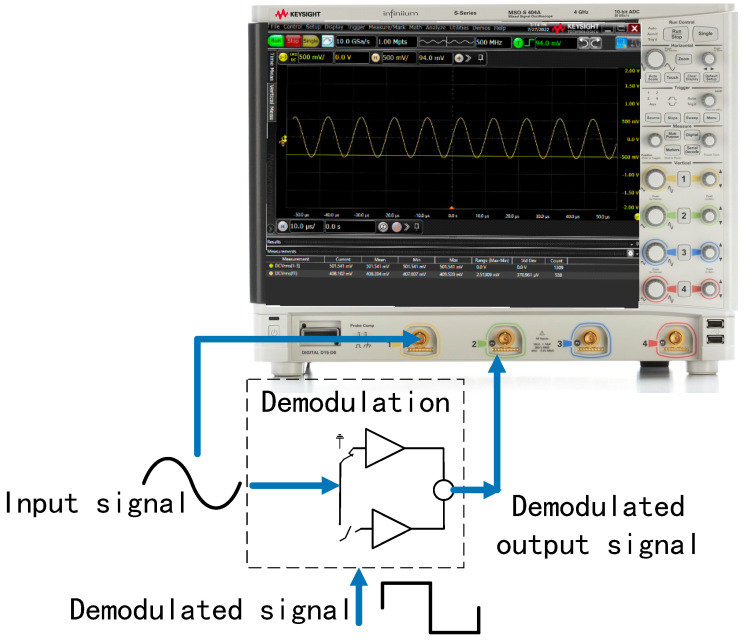
Test connection diagram of demodulation circuit.

**Figure 14 sensors-22-09284-f014:**
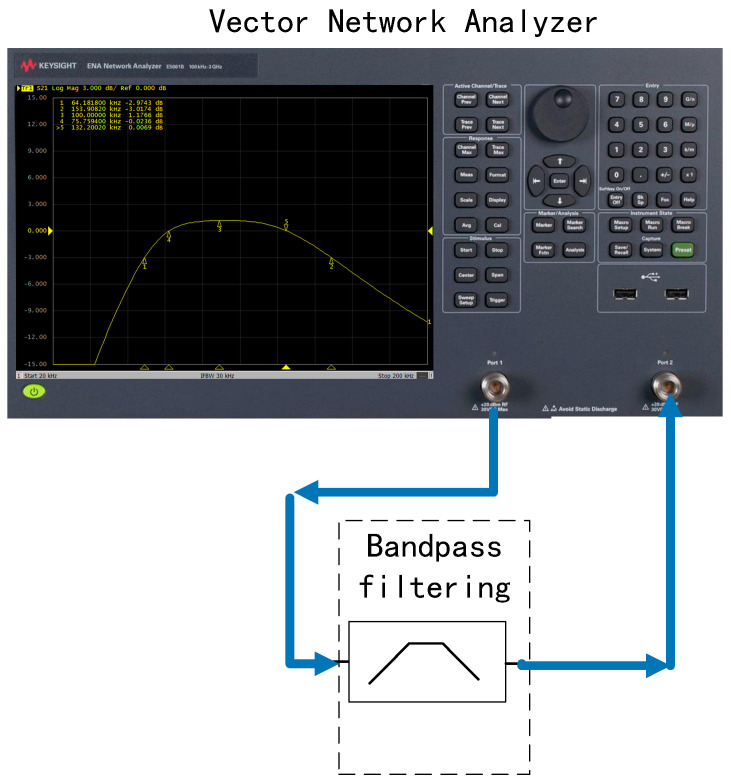
Band-pass filter circuit test connection diagram.

**Figure 15 sensors-22-09284-f015:**
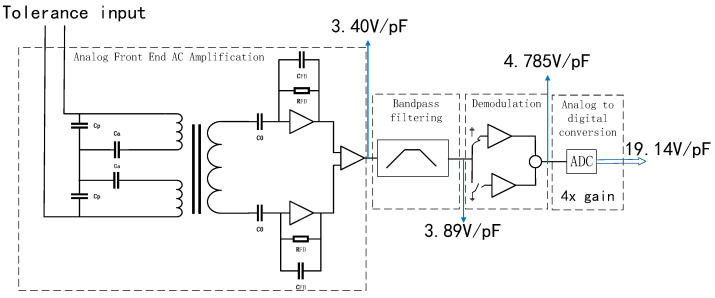
Gains of each functional module of the capacitive sensing system.

**Figure 16 sensors-22-09284-f016:**
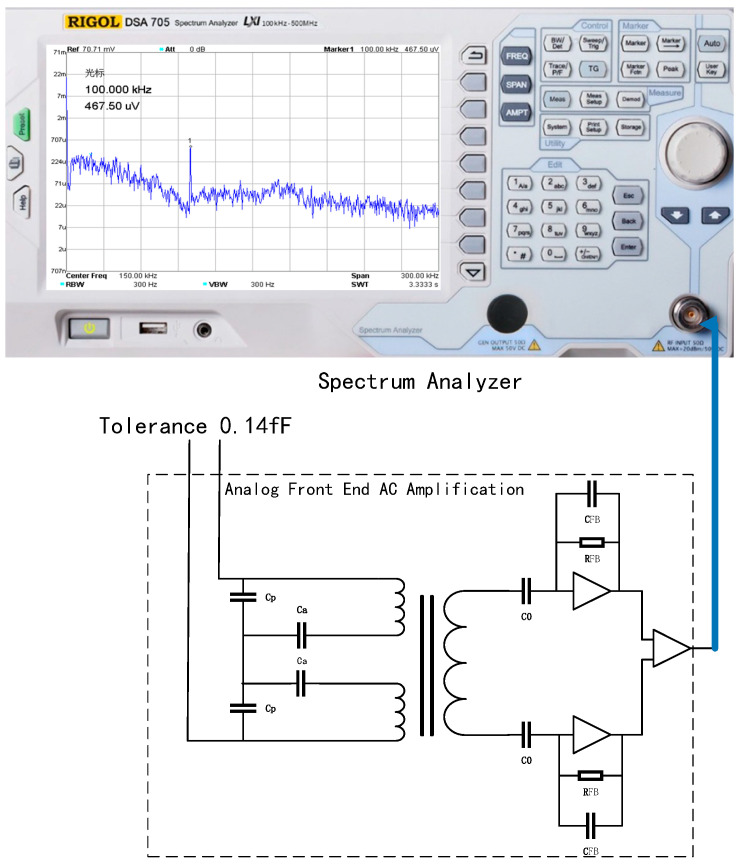
Capacitive sensing system demodulation function module under measurement 0.14 fF tolerance.

**Figure 17 sensors-22-09284-f017:**
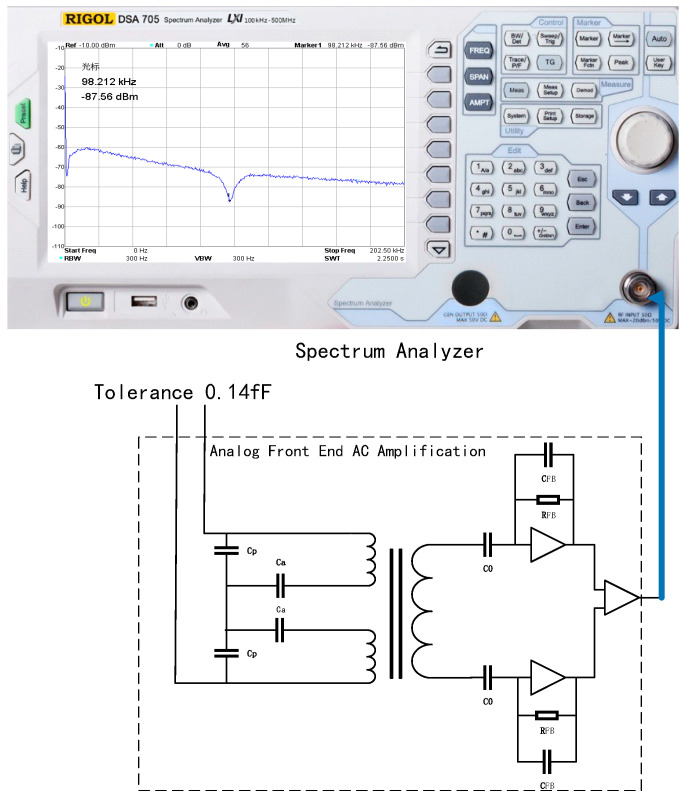
Capacitive sensing system demodulation function module under zero-tolerance measurement.

**Figure 18 sensors-22-09284-f018:**
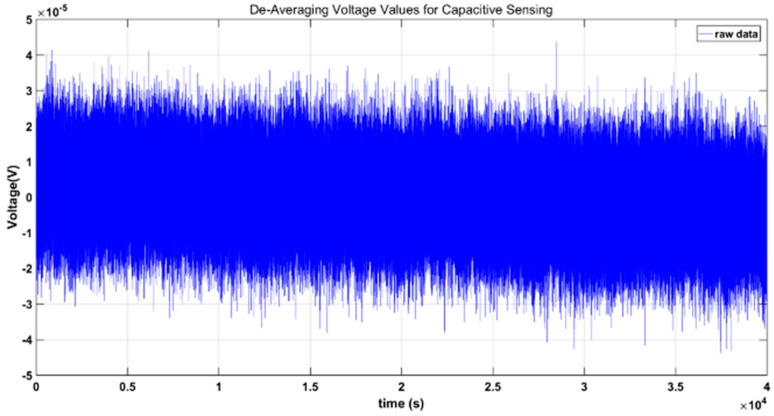
De-averaging the voltage values for capacitive sensing.

**Figure 19 sensors-22-09284-f019:**
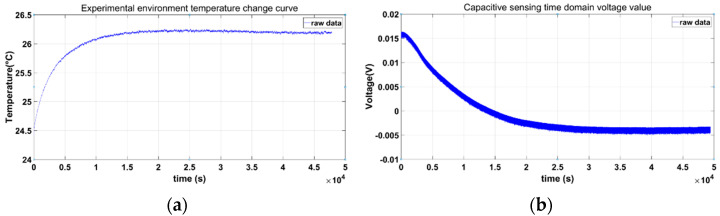
(**a**) Experimental environment temperature change curve. (**b**) Capacitive sensing time-domain voltage value.

**Figure 20 sensors-22-09284-f020:**
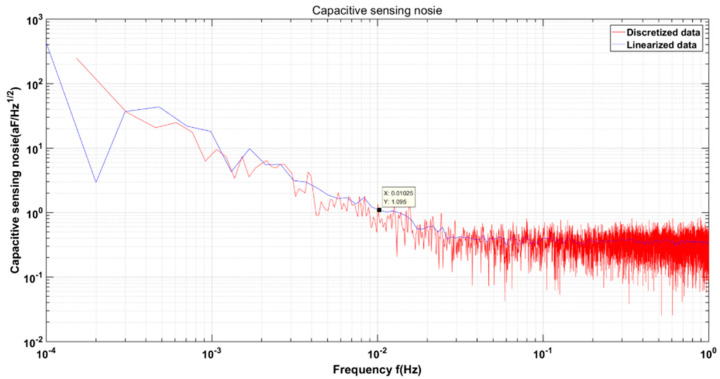
Capacitive sensing noise.

**Table 1 sensors-22-09284-t001:** Parameters of the third-order band-pass filter.

Parameter	Gain at 100 kHz	Lower −3 dB Frequency	Higher −3 dB Frequency
2nd order low-pass	1.7		140 kHz
2nd order high-pass	0.7	71 kHz	pF
1st order band-pass	0.82	26 Hz	409 kHz
Output driver	1.02		9.6 MHz
Combined filter	1	71 kHz	139 kHz

**Table 2 sensors-22-09284-t002:** Electronic design parameters of capacitive sensing system.

Parameter	Numerical Value	Unit
Excitation signal frequency ($\omega_{0}$)	100	kHz
Scientific range tolerance ($\Delta C_{max}$)	0.12	pF
Capacitive sensing DC voltage acquisition range	2.5	V
Tuning capacitor ($C_{p}$)	22	nF
Capacitance at the center of the TM electrode ($C_{0}$)	1.15	pF
Planar transformer inductance value (*L*)	4.2	mH
Planar transformer quality factor (Q)	180	

**Table 3 sensors-22-09284-t003:** Sensitive structural parameters.

Parameter	Numerical Value	Unit
Test mass weight	107	g
Test mass size	46 × 46 × 46	mm
*X*-axis plate area	522	${\mathrm{mm}}^{2}$
Initial spacing of *X*-axis polar plates	4	mm
*Y*-axis plate area	271.22	${\mathrm{mm}}^{2}$
The initial spacing of the *Y*-axis polar plates	2.9	mm
*Z*-axis plate area	240.5	${\mathrm{mm}}^{2}$
*Z*-axis polar plate initial spacing	3.5	mm

**Table 4 sensors-22-09284-t004:** Sensitive structure: six pairs of electrode capacitance tolerance and capacitance sensing system readout voltage.

Electrode Pair	Tolerance Value (pF)	Voltage Value V
X. A+ and X. A−	0.0319	0.61024
X. B+ and X. B−	0.0050	0.09571
Y. A+ and Y. A−	0.0443	0.84834
Y. B+ and Y. B−	−0.0021	−0.04019
Z. A+ and Z. A−	0.0122	0.23363
Z. B+ and Z. B−	0.0175	0.33495

**Table 5 sensors-22-09284-t005:** Comparison of capacitive sensing noise resolution.

Research and Development Organization	$\mathrm{Capacitive}\;\mathrm{Sensing}\;\mathrm{Noise}\;(aF/\sqrt{Hz}@10\;mHz$)	Supplementary Notes
Huazhong University of Science and Technology, China [9]	0.2	$\mathrm{Published}\;\mathrm{in}\;2017.\;\mathrm{Taking}\;\mathrm{an}\;\mathrm{accelerometer}\;\mathrm{with}\;a\;\mathrm{design}\;\mathrm{resolution}\;\mathrm{of}\;2\times 10^{-12\;}{\mathrm{ms}}^{-2}/\sqrt{\mathrm{Hz}}$.
Lanzhou Institute of Technology and Physics, China	1.27	Developed in 2021. Unpublished.
LISA Pathfinder [7]	0.7–1.8	In-flight measurements. Published in 2017.
Changchun Institute of Optics, Fine Mechanics and Physics, China	1.095	$\mathrm{Closed-loop}\;\mathrm{measurements}\;\mathrm{of}\;\mathrm{capacitive}\;\mathrm{sensing}\;\mathrm{circuits}\;\mathrm{and}\;\mathrm{sensitive}\;\mathrm{structures}\;\mathrm{of}\;\mathrm{inertial}\;\mathrm{sensor}.\;\mathrm{The}\;\mathrm{acceleration}\;\mathrm{noise}\;\mathrm{target}\;\mathrm{is}\;3\times 10^{-15}{\;\mathrm{ms}}^{-2}/\sqrt{\mathrm{Hz}}$.

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
