# Peer review of "Analysis of a Capacitive Sensing Circuit and Sensitive Structure Based on a Low-Temperature-Drift Planar Transformer"

_sensors, 2022, doi:10.3390/s22239284_

Round 1
Reviewer 1 Report
Please see the attachment.

Reviewer 2 Report
9-10 unclear sentence
In the space gravitational wave detection mission, the inertial sensor is used as the core load, and its acceleration noise
needs to reach 3 × 10−15??−2/√?? at a frequency of 0.1???, which corresponds to the capacitive sensing system (??)The capacitive
sensing noise on the sensitive axis needs to reach 1??/√??.
65: need a dot
. The minumum...
from line 65 to line 68: unclear sentence.
What do you want to say?
Maybe something like this?
"Based on the requirements of China's future space gravitational wave detection Taiji program,
for the low-frequency high-precision resonant bridge circuit, a planar transformer with high Q value,
low-temperature drift, and low 1/? noise capacitive sensing circuit, and closed-loop measurement of sensitive structures are developed.
The minimum capacitance resolution in the time domain is about 3??, which is much lower than the scientific measurement resolution requirement of 5.8??.
For gravitational wave detection, in the frequency band of 10??? − 1??, the capacitive sensing noise is measured to be 1.095??/√??."
from line 80 to line 83: This is the conclusion of the previous paragraph. Maybe you should write an appropriate introduction for this section.
"The content of this article is as follows. The second section mainly introduces the design of the planar transformer, 80
the composition and basic working principle of the capacitive sensing circuit. Section III presents the ground experi- 81
mental setup and actual measurement and data evaluation results. The final section provides a comprehensive sum- 82
mary of the work of this document"
177: the B of the board should be a capital letter
195:
...amplitude remains unchanged. The bandwidth range is 71kHz~139kHz.
203: Maybe you miss an 'and'
'The modulated signal first needs to go through the electronic switch demodulation circuit to separate the 100???
carrier signal from the noise spectrum (??and) to separate the voltage signal close to DC'.
301:Maybe you don't need a dot
required for gravitational wave detection.(??) Scientific measurement resolution.
Fig 12, 19: You should use a different and big font for the label of the figures. It is difficult to read the unit and the number on the axes.
About the conclusions: Looks similar to the introduction.
You should rephrase that.that.
Reviewer 3 Report
The paper presents a capacitive circuit using a planar transformer with low temperature drift. The authors described mainly introduces the design of the planar transformer, the composition and basic working principle of the capacitive sensing circuit.
The paper is very interesting from the metrological point of view, theoretical and practical.
The paper has good scientific level and is written in good language, but some of the sentences are too long. Therefore some parts of this paper are difficult to understand. I recommend to take this fact into account in future publications.
The list of References is sufficient.
Below, a few remarks were presented. Please take them into consideration in final version of this article.
- Line 96, instead of the designation Figure 3 shouldn't it be Figure 2?
- The beginning of the sentence should be from a capital letter.
- Line 184 the reference to Figure 6 should be before the figure, not after it.
- Line 207, I propose to change the abbreviation op amp, operational amplifier, to OP and then use this abbreviation in the rest of the article.
- Line 216 remove the space between the word Figure 9 and the dot.
- Figure 10 and figure caption are on pages 8 and 9, this should be changed.
- Table 3 is divided into 2 pages. The table should be on one page.
- Chapter 4, Measurement result and discussion, I suggest moving to the next page.
- Line 109, instead of the unit 100 KHz it should be 100 kHz.
- There is no reference to Figure 8 in the text of the paper.
- Axis descriptions in Figures 5, 12 and 19 are barely legible.
- What does the uV unit, lines 284, 291, 299, mean?
Round 2
Reviewer 1 Report
In the revised manuscript, when comparing the capacitive sensing noise resolution, the authors missed one literature [Research and Development of Electrostatic Accelerometers for Space Science Missions at HUST, Sensors 17(9):1943 (2017)], in which the capacitive sensing noise is about 0.2 aF/rt(Hz) @10mHz.
Author Response
“请参阅附件。
